# Rapid Identification of Infectious Pathogens at the Single-Cell Level via Combining Hyperspectral Microscopic Images and Deep Learning

**DOI:** 10.3390/cells12030379

**Published:** 2023-01-19

**Authors:** Chenglong Tao, Jian Du, Junjie Wang, Bingliang Hu, Zhoufeng Zhang

**Affiliations:** 1Xi’an Institute of Optics and Precision Mechanics, Chinese Academy of Sciences, Xi’an 710119, China; 2University of Chinese Academy of Sciences, Beijing 100049, China; 3Key Laboratory of Biomedical Spectroscopy of Xi’an, Xi’an 710119, China

**Keywords:** infectious pathogens, species identification, hyperspectral microscopic imaging, transfer learning, spectral characteristics

## Abstract

Identifying infectious pathogens quickly and accurately is significant for patients and doctors. Identifying single bacterial strains is significant in eliminating culture and speeding up diagnosis. We present an advanced optical method for the rapid detection of infectious (including common and uncommon) pathogens by combining hyperspectral microscopic imaging and deep learning. To acquire more information regarding the pathogens, we developed a hyperspectral microscopic imaging system with a wide wavelength range and fine spectral resolution. Furthermore, an end-to-end deep learning network based on feature fusion, called BI-Net, was designed to extract the species-dependent features encoded in cell-level hyperspectral images as the fingerprints for species differentiation. After being trained based on a large-scale dataset that we built to identify common pathogens, BI-Net was used to classify uncommon pathogens via transfer learning. An extensive analysis demonstrated that BI-Net was able to learn species-dependent characteristics, with the classification accuracy and Kappa coefficients being 92% and 0.92, respectively, for both common and uncommon species. Our method outperformed state-of-the-art methods by a large margin and its excellent performance demonstrates its excellent potential in clinical practice.

## 1. Introduction

Infectious diseases have been plaguing human health and threatening human lives [1,2,3,4]. How to detect pathogens quickly and accurately has long been a problem. Currently, the fastest method by which to precisely detect pathogens is matrix-assisted laser desorption ionization–time of flight (MALDI-TOF) [5,6,7], which is based on species-specific spectra of peptides and protein masses obtained by mass spectrometry. However, MALDI-TOF still requires bacteria to proliferate to more than 100 thousand [8,9], which dramatically slows down the detection. Another method that allows the rapid identification of pathogens is polymerase chain reaction (PCR); however, its high sensitivity and specificity come at the cost of high prices and complexity, which limits the application and popularization of this technique in clinical practice. Thus, an inexpensive and rapid method for detecting pathogens without requiring bacterial proliferation is highly desirable. How to identify individual bacterial species quickly, accurately, and cheaply is a crucial question.

Hyperspectral microscopic imaging (HMI), as a non-invasive, chemical-free, and label-free imaging technique, can simultaneously capture hundreds of bands of images at the bacterial level to form a hyperspectral microscopic image [10,11], also known as a microscopic datacube (MD). The biological features of pathogens, reflected by spectral and morphological information, are encoded into MDs [12]. The key to detecting bacteria is to encode more and finer biological features into MDs and extract the robust species-dependent features [13,14].

Recently, several researchers have used MDs to classify pathogens [15,16,17,18]. Nevertheless, the narrow wavelength range and coarse spectral resolution MDs contained insufficient biological information for the more complex detection of infectious pathogens [19,20,21,22,23,24,25,26,27]. The low utilization of the data further compounded the lack of information due to the use of only the spectral profile of one or the average pixel. Park et al. input the spectral profile directly into machine learning algorithms for bacterial classification, which was not robust due to the lack of high-level features [19,23,28]. Kang et al. extracted high-level features of the spectral profile using deep learning [19,23,29]. However, the extracted features lacked morphological information. Additionally, their small-scale dataset could not reflect the actual data distribution of pathogens and thus led to less robust performance for clinical diagnostic scenarios that require high stability.

In our previous research, we achieved the goal of classifying genera with large-scale data with a finer spectral resolution and a wider wavelength range [30]. However, the identification of bacteria at the species level is more relevant for clinical diagnosis because the prescribed antibiotics vary depending on the species detected. Because species are a more refined division of the genus, there are fewer biochemical differences among species and further types, making it more difficult to classify species in comparison to genera. Additionally, how to identify uncommon pathogens is also a problem resulting from the imbalanced number of samples of common and uncommon pathogens.

In order to rapidly identify species of infectious pathogens (including common and uncommon ones) at the single-cell level so as to avoid the need for time-consuming culture and staining of bacteria, it is necessary to collect more data and develop more advanced algorithms.

In terms of data, an MD dataset with an enormous amount of data labeled for species has been built. Compared with our previous research on identifying genera, we enlarged the types of pathogens from 11 to 22 and increased the number from 110,000 to 170,000. In terms of algorithms, we developed an end-to-end bacteria identification network (BI-Net) to automatically extract the biometric features encoded in the MDs. To enhance the feature representation of the BI-Net, we fused low-level features containing abundant details with high-level features containing semantic information. Furthermore, we used a three-dimensional convolutional neural network (3D-CNN) as the convolutional kernel of BI-Net to extract spatial–spectral features [31]. After analyzing the features extracted by BI-Net, we proved that common pathogens could be accurately identified by combining HMI and deep learning.

It is a natural idea to use large-scale data to predict the data distribution of common pathogens and thus their identification [32,33]. However, due to the small number of uncommon pathogens, how to identify them is also an unsolved problem [34,35]. In contrast, the deep learning method driven by large-scale data is unsuitable for uncommon pathogens. Therefore, we transferred the knowledge of common pathogens obtained from BI-Net to rare pathogens, achieving excellent recognition accuracy [36].

## 2. Materials and Methods

### 2.1. Preparation of Bacteria

A total of 22 species including 62 strains from 16 sources, including strains from the American Type Culture Collection (ATCC), which contained 15 types of body fluid, were collected from The Second Affiliated Hospital of Air Force Military Medical University from July to November in 2021. In addition to clinical strains, strains from the ATCC were also used in the study. To enhance the applicability of adapting our method to complex clinical diagnostic scenarios, we made every effort to enlarge the pathogen sources. The pathogens involved in our experiments stemmed from 16 sources, i.e., urine, wound secretions, sputum, tissue fluid, articular cavity fluid, ascites, cerebrospinal fluid, purulence, lavage fluid, punctured fluid, drainage fluid, catheter, blood, and the ATCC.

All the bacteria were cultured to produce colonies on blood plates for 24–48 h in incubators at 35–37 °C. A pure isolated colony was dissolved into saline in a high-pressure sterilized turbidity tube for 30 min. To obtain the MDs of individual bacteria as samples, bacterial suspensions of pure isolated colonies were spun in a centrifugation machine at 3000 rpm for 10 min or 5000 rpm for 5 min to separate the pathogens. Subsequently, the suspension was dripped and coated on a glass slide, and NaCl crystals were dissolved with 75% alcohol to facilitate observation, imaging, and analysis.

### 2.2. The Collection and Preprocessing of MDs

#### 2.2.1. The HMI System

The self-assembled HMI system had two components. One was a hyperspectral sensor (acA2000-165 umNIR, Basler Ahrensburg, Germany) enhanced in the near-infrared wavelength range. The other was an upright microscope (BX43, Olympus, Tokyo, Japan), whose original light source was replaced by a 30 W halogen lamp to cover the wide wavelength range. In order to obtain more detailed spatial information, we used MPlanFL N (100×/0.90) as an objective lens with over 65% transmittance in the valid wavelength range. Therefore, the lens had adequate coatings to cover a wide spectral range. Owing to the advanced equipment, we obtained MDs with a wide wavelength range (440–1023 nm), a high optical resolution (0.4 um @600nm wavelength), a high spectral resolution (2.1 nm), and a large field of view (64 × 64 um^2^), which enabled us to detect species of infectious pathogens.

#### 2.2.2. The MD Collection for Pathogens

A self-developed HMI system (Figure 1A) was used to image the pathogens. After about 30 s of scanning, an MD of 1000 × 1000 × 277 was obtained (Figure 1B). The image resolution was 1000 × 1000, while the number of wavebands within the range of 440–1023 nm was 277. The MD contained rich spectral (Figure 1D) and morphological information (Figure 1E). The spectral information in the MD of individual bacteria (Figure 1C,D) was able to reflect the biochemical composition of the bacteria, which is key to identifying the strain. Further, morphological information can respond to the association of spectral information among bacterial localities and therefore is also essential in identifying species.

#### 2.2.3. The Preprocessing of MDs

The raw MDs had 200 bands after abandoning the front and rear 77 bands with a lower signal-to-noise ratio. The shape of the valid MDs was 1000 × 1000 × 200. As we aimed to recognize the pathogen as fast as possible and eliminate the culture, we needed to obtain the MD of a single bacterium. However, there was more than one bacterium in the original MD that the HMI system captured. To address this issue, we cropped an individual bacterium from the original image (Figure 2C). Before this, preprocessing was executed on the raw datacube. The following is the workflow of the whole process. After the raw MDs were collected, flat-field correction was used to correct uneven brightness among rows [37].
(1)$$M_{r,c}^{even}=\frac{M_{r,c}^{raw}}{\frac{1}{n}\times \sum_{k}^{n}M_{k,c}^{raw}}$$

In Equation (1), $M_{r,c}^{even}$ is the spectral profile at the position (r, c) of the corrected MD and $M_{r,c}^{raw}$ of the raw MD. In order to avoid being affected by outliers, the maximum and minimum 150 spectral values in a column were discarded, and n is the number of remaining pixels. A binary image $\;Mask^{bac}$ with a possible individual bacterium and a background $Mask^{bg}$ were obtained using K-means [38]. It was observed that the foreground in $Mask^{bac}$ did not contain all the bacteria. Those too small and too large were impurities and unclassified bacteria, respectively. Impurities, bacterial colonies, and incomplete bacteria on the image boundary were discarded, and the remaining single bacteria were retained as samples. The average spectral curve of the background, SC, was then obtained according to Equation (2), where subscript b denotes the bth band, and ∗ is Hadamard product.
(2)$$SC_{b}=\frac{\sum Mask^{bg}*MD_{b}^{even}}{\sum Mask^{bg}}$$

Since there is no interference, SC can reflect the intensity of illumination. To eliminate the influence of different light source brightness, transmittance MD ($MD^{tran}$) was obtained by dividing the $MD^{even}$ with SC band-wise.
(3)$$MD_{r,c,b}^{tran}=\frac{MD_{r,c,b}^{even}}{{\mathrm{SC}}_{b}}$$

Finally, MDs for 22 species were collected and were divided into two groups: common species and uncommon species. More specifically, we collected 15 common species with more than 170,000 MDs and seven uncommon species with more than 2100 MDs. Finally, we split the database of the common species using a ratio of 7:3 for training and testing, but a ratio of 3:7 for the uncommon species to validate the performance of transfer learning.

### 2.3. Algorithm Design

#### 2.3.1. BI-Net

Compared with conventional RGB images, the two most predominant features of MDs are higher dimensions (the number of channels) and fewer pixels. Therefore, we proposed a network specifically tailored to classifying the MDs of bacteria, called BI-Net, as shown in Figure 2D. The following are the reasons for designing the network. Firstly, because the high dimensions of MDs mean that a large amount of spectral information can be obtained, the 3D-CNN was adopted to simultaneously extract the spatial and spectral information in the MDs. As in Equation (4), a 3D signal g is mapped to another signal f through a 3D filter h. f still retains the 3D information, which is significant for extracting spectral–spatial features for the detection of species.
(4)$$f_{r,c,b}={\left(h*g\right)}_{r,c,b}=\sum_{i=1}^{t}\sum_{j=1}^{t}\sum_{k=1}^{t}h_{r,c,b}\times g_{r-i,c-j,b-k}$$

Moreover, the feature fusion was applied to augment the representative capability of the feature map of the bacteria because bacteria are difficult to detect due to their small number of pixels (mostly less than 32 × 32) [39].
(5)$$F=Concat(pooling(f_{i})),i=1,2,3,4$$

As shown in Equation (5), for four residual blocks, the feature maps ($f_{i}$, i=1,2,3,4) outputted by each residual block were pulled into a vector and then concatenated (Concat) together as in Equation (1). After feature fusion, the feature F (1 × 4608) was then transmitted to the fully connected layer, and the probability for every species was obtained by the SoftMax layer.
(6)$$SoftMax(Z)=\frac{e^{z_{i}}}{\sum_{k=1}^{N}e^{z_{k}}},i=1,2,\ldots ,N$$

In addition, a three-dimensional average pooling based on four different pooling kernels was adopted to pool the four feature maps, which were outputted by the four residual blocks to lower the feature dimension of the superficial layers so as to prevent the classifier from being largely affected by the superficial features. The sizes of the pooling kernels were [10,5,5], [8,3,3], [5,3,3], and [2,2,2]; and the feature dimensions were 1 × 1024, 1 × 1536, 1 × 512, and 1 × 1536, respectively. Finally, the ResNet connection was also used to avoid a vanishing gradient and an exploding gradient in the deep network [40]. Additionally, each convolutional layer was followed by a batch normalization (BN) so as to accelerate the network learning speed [41].

#### 2.3.2. Transfer Learning

The identification of common species was performed using the trained BI-Net on the dataset. However, the small number of images made it impossible to train the BI-Net to identify uncommon species. Therefore, transfer learning was adopted to finetune the parameters, transferring the knowledge from the source domain with a larger sample size to the target domain. In this study, the source domain referred to the 15 common species, and the target domain to the seven uncommon species. The trained parameter in the source domain was finetuned and the classifier, which was located after the fully connected layer of the BI-Net, was redesigned. More specifically, the only fully connected layer in the classifier was changed to three (Figure 2E). In addition to the last layer, every fully connected layer was followed by Relu to strengthen the nonlinear transformation. The transfer learning and the redesigned classifier are shown in Figure 2E.

### 2.4. Super Parameters for Training

We performed many experiments to determine the super parameters to obtain a satisfactory result. Listed in Table 1 are the parameters that produced the optimal results.

## 3. Results

The flow chart of our methodological framework is shown in Figure 2. The samples were collected and grouped into common and uncommon pathogens. The designed BI-Net was used to identify the common pathogens, and the knowledge acquired was then transferred to the uncommon pathogens.

Firstly, we developed an HMI system with a broad range of wavelengths and a fine spectral resolution. After bacteria in the microscopic field using a view at 100× magnification were imaged by the HMI system, an original micro-datacube was then acquired. In order to avoid the time-consuming incubation of pathogens for identification, we needed to detect bacterial species at the level of individual bacteria. Therefore, we cropped individual bacterium taken as a sample from the original micro-datacube. 

All of the isolates that we used in the study were from clinical specimens. In addition, bacteria from the American Type Culture Collection (ATCC) were also used to diversify our data. To enhance the soundness of adapting our method to complex clinical diagnostic scenarios, we made every effort to enlarge the pathogen sources. The sources involved in our experiments stemmed from urine, wound secretions, sputum, tissue fluid, articular cavity fluid, ascites, cerebrospinal fluid, purulence, lavage fluid, punctured fluid, drainage fluid, catheter, blood, and the ATCC. Ultimately, our experiment contained 22 species and 62 strains, built as a large-scale, multi-source MD dataset of infectious pathogens.

The network structure for deep learning is shown in Figure 2E. Large-scale MDs of individual bacteria were input into the network for training. The network continuously adjusted the parameters to reduce errors by back-propagating so as to accurately predict unseen sample species. After being trained with common pathogens, BI-Net obtained the knowledge required to extract high-level features for the identification of species. However, training based on large-scale data for uncommon pathogens is not practical. The powerful capability of the network to learn features from small amounts of data would result in overfitting. In order to perform accurate identification of uncommon pathogens, we transformed the knowledge obtained regarding the identification of common pathogens for use in the identification of uncommon ones.

This section presents the results in the following logical order: First, we analyzed the spectral characteristics of 22 species to explore their intrinsic connections. After confirming that the spectral profiles differed, we investigated whether relying on the spectral characteristics to perform species identification was sufficient. As a result, the micro-datacubes were found to significantly improve the recognition results as compared to the spectral profiles. However, we recognized that directly training the network to identify uncommon pathogens was not desirable. The following experiment was carried out to test whether transfer learning could be used to identify uncommon pathogens based on the knowledge obtained for the identification of common pathogens. Afterward, we explored the performance of the identification of both common and uncommon pathogens and found that our approach worked quite well. To explore the specific advantages of our method, we visualized and analyzed the raw spectral profiles and the high-level features extracted by deep learning.

### 3.1. The Spectral Analysis of Pathogens

Since differences in the material composition of bacteria can be distinguished by their spectral characteristics, it was necessary to analyze the spectral profiles using large-scale MDs of pathogens to explore the intrinsic connections. The average spectral profiles of all the samples were statistically analyzed and visualized (Figure 3). For ease of observation and comparison, species belonging to the same genus were placed together in a column and visualized with the same color. The spectral profiles were processed to eliminate the effect of the light source by dividing the average spectral value of the background.

As shown in Figure 3, on the valid bands between 547 nm and 969 nm, all the species essentially trended downward and then upward, which indicates that less or more light penetrated the pathogens, respectively. 

In addition, the magnitude of the decline and rise varied considerably among species. Some pathogens were relatively smooth, such as *P. aeruginosa* (Figure 3L) and *E. coli* (Figure 3T), while others had significant ups and downs, such as *S. aureus* (Figure 3D) and *S. saliva* (Figure 3M). The differences of the minimum and maximum values for *P. aeruginosa* and *E. coli* were relatively small; on the other hand, the range of the minimum and maximum values for *S. aureus* was relatively wide. The mean and minimum values of Enterococcus species, including *E. faecium* (Figure 3I) and *E. faecalis* (Figure 3L), were essentially the same.

As for similarities, we noticed that different species of the same genus had varying spectral curves. Nevertheless, they still had relatively similar spectral profiles. While the lower and average curves of *S. haemolyticus* (Figure 3C) and *E. faecium* (Figure 3I) were similar, the upper bound of the spectral curves differed. 

### 3.2. Microscopic Datacube Classification Analysis

Differences between the spectral characteristics of species are crucial to identifying pathogens. However, we were uncertain whether using spectral profiles to detect pathogens was optimal, so a total of five data types were used for classification comparison as follows: 1. spectral profile: the average spectral profile of an individual bacterium; 2. binary mask: containing most morphological information of the bacterium such as area, perimeter, and aspect ratio; 3. grey image: single-band images simulating ordinary optical imaging; 4. pseudo-color image: three-channel images simulating ordinary optical imaging; and 5. microscopic datacube: containing not only abundant spectral characteristics, but also spatial information.

As shown in Figure 4A, the binary mask had the lowest accuracy of 27%, indicating that using only morphological information to identify the species of bacteria is not sufficient. Compared to the binary mask, the grey image had richer color information and its accuracy was more than twice that of the binary mask, at 56%. Since the information in the three channels of the pseudo-color image was higher than that in the binary and greyscale maps, it is not surprising that it had an accuracy of 66%.

Furthermore, we observed that the classification accuracy of the spectral profile was 85%, much higher than that of the above three data types. It was noted that the spectral profile contained no morphological information regarding the bacteria; however, it achieved an impressive accuracy, demonstrating the capability of spectral characteristics to identify minor biochemical differences among species. The highest accuracy (92%) was achieved using MD which encoded the morphological characteristics of the bacteria in addition to the spectral features. 

The transmittance of images at different wavelengths and locations for the pathogens was different, which thus led to the difference in the spatial and spectral information that more specifically showed the differences between the morphology and biochemistry of the species. This is the reason for the increase in accuracy (from 85 to 92%). Therefore, according to the analysis, the use of MDs is more suitable for the identification of species than spectral profiles. 

### 3.3. Transfer Learning Performance Comparison

After determining that MDs were the most suitable data type for bacterial identification, we used them to identify pathogens. However, due to the lack of samples of uncommon pathogens, direct training led to poor classification performance (Figure 4B). Given the successful application of transfer learning, we considered uncommon pathogens as a target domain and transferred the knowledge from the source domain (common pathogens) to the identification of uncommon pathogens. Nevertheless, the question arises as to whether the transfer learning worked and in what way. To answer these two questions, we experimented by comparing direct training and transfer learning. To better approximate the data distribution of the uncommon pathogens, we redesigned the classifier for the uncommon pathogens. Only one fully connected layer was used for common pathogen species, while three fully connected layers were used for uncommon pathogen identification. Additionally, to better fit the nonlinear variation, we executed the ReLu (rectified linear units) activation function after every fully connected layer except for the last one.

The data, classifiers, and super parameters were kept constant throughout the experiment to ensure fairness. The difference was that direct training thoroughly trained the entire network from random initialization, while transfer learning froze or finetuned the trained backbone weights for common pathogens and then trained the classifiers. In comparison to direct training, the accuracies of freezing and finetuning were 24.5% and 28.7% higher, with Kappa coefficients of 0.30 and 0.34 greater (Figure 4B). This remarkable result indicates that the generalization ability of transfer learning is much stronger than that of direct learning. The knowledge obtained regarding common pathogens played a significant role in identifying uncommon pathogens. The finetuning weights had a 3.2% increase in accuracy, and the Kappa coefficient was 0.04 higher than that of the fixed weight approach. The high accuracy and Kappa coefficient of the finetuning weights were why we chose this method for identifying uncommon pathogens. 

Given the above, the experiments illustrate the outstanding usefulness of transfer learning in identifying common pathogens and the necessity to use transfer learning for uncommon pathogens. 

### 3.4. Pathogen Identification Using MDs and Deep Learning

Having determined the data types and classification methods, we explored the effectiveness of identifying common and uncommon pathogens.

#### 3.4.1. Identification of Common Pathogens

We used the deep learning network BI-Net to identify 15 species of common pathogens, with an overall recognition accuracy of 92% and a Kappa coefficient of 0.92. The recognition accuracy for each species is presented in a confusion matrix in Figure 5A. Among these, the recognition accuracy for *K. pneumoniae, A. baumannii*, and *S. marcescens* were the lowest among the 15 species, at 88%. All the other species had a precision of 90% or more, with the highest recognition accuracy of 97% for *S. aures*, *P. vulgaris*, *P. aeruginosa*, and *M. morganii*. The values below 1% are not shown in the confusion matrix.

*S. haemolyticus* had a 7% probability of being misclassified as *E. faecium*, with a corresponding 6% of *E. faecium* being identified as *S. haemolyticus*, which is ascribed to their similar spectral profiles. The probability of misclassification between all the other species, even if they belong to the same genus, was less than 4%, which is consistent with the fact that the difference between their spectral curves was not significant enough.

The receiver operating characteristic (ROC) curve was used to evaluate the performance, and the area under the curve (AUC) positively correlated with the specificity but negatively with the sensitivity. As shown in Figure 5B, the AUC scores for all the pathogens were higher than 0.99. This indicates that our algorithm made accurate predictions for all the species.

#### 3.4.2. Identification of Uncommon Pathogens

Using BI-Net trained with common pathogens, the backbone weights were finetuned, and the classifier was redesigned and trained for the detection of uncommon pathogens. Out of more than 2000 samples, 30% were used for training and 70% for testing. The overall recognition accuracy for common pathogens was 92%, with a Kappa coefficient of 0.92. The classification confusion matrix and ROC curves are shown in Figure 5C,D.

The transformed model had 98% accuracy for *S. anginosus*, in comparison with only 78% for *M. catarrhalis*. The probability of misclassifying this species as any other species was higher than 1%. The high probability of misclassification was due to the broad spectral coverage of *M.catarrhalis* that rendered a single bacterium highly likely to resemble other species.

The probability of misclassification of species belonging to Streptococcus was negligible. However, despite the different spectral profiles of Staphylococcus species, 6% of *S. warneri* were misclassified as *S. saprophyticus* and 9% of *S. saprophyticus* as *S. warneri*. This phenomenon may have been caused by the number of uncommon species being so small that statistical significance was not apparent.

Based on the ROC curve (Figure 5D), the AUC scores of the uncommon pathogens exceeded 0.98, indicating that transfer learning was able to effectively address sample imbalance and accurately identify uncommon pathogens.

### 3.5. Representational Learning Capabilities for Deep Learning

To investigate why our method achieved such an excellent performance, we visualized and compared the spectral profiles of pathogens with the high-level features extracted by BI-Net (Figure 6A,D). The visualization method used was t-SNE (t-distributed stochastic neighbor embedding) [42], which clusters samples in a low-dimensional space by dimensionally reducing the data and maintaining invariance in the distance between samples to respond to the classification effect of features.

As shown in Figure 6A, although some small clusters emerged, the individual categories were scattered in a confusing cluster and failed to form category-defined clusters. This indicates that the direct use of the low-level features of the spectral curve for classification was not appropriate. In contrast, an evident clustering phenomenon emerged after visualization of the high-level features extracted by BI-Net (Figure 6B). Only a small number of samples were projected into clusters of other species.

Further, considering the high dimensionality (4096) of the extracted features and the large volume of data (over 170,000), a small number of misclassifications in the visualization plot is understandable and acceptable. Importantly, it can be observed that the features extracted by BI-Net were able to classify the species well. The visualization comparison results show that even though there were different spectral profiles, it is not reasonable or desirable to use original spectral features directly for classification. In order to identify bacteria, high-level biochemical characteristics needed to be extracted. The satisfactory clustering results after extraction indicate that BI-Net learned the species-dependent characteristic of the MDs.

Due to the small sample size of the uncommon species (less than 2300), the features extracted by transfer learning did not form very distinct clusters after the t-SNE visualization (Figure 6C,D). The samples for species were mainly distributed randomly across the plane (Figure 6C). However, the extracted high-level features of each species leaned towards each other (Figure 6D). This indicates that transfer learning extracted high-level features conducive to species classification, and that the approach of transferring knowledge obtained regarding common pathogens to uncommon pathogens is feasible.

### 3.6. Comparison with State-of-the-Art

BI-Net was quantitatively compared with several state-of-the-art algorithms to further demonstrate its performance. These methods were ResNet [40] and DenseNet [43], whose performance is excellent in image classification. Furthermore, since 1D-CNN [25] and Fusion-Net [24] have outstanding performance in hyperspectral image classification, we also chose these as the algorithms for comparison. For all the comparison methods, the training and testing data were set to be the same.

Table 2 shows the accuracy and Kappa coefficients of all the methods. It can be seen from the data in the table that both the accuracy and Kappa coefficient of BI-Net were better than those of the other classifiers. In other words, the proposed method performed favorably against the other state-of-the-art methods.

## 4. Discussion

In this study, we constructed a multi-source, large-scale species identification dataset using the developed HMI system. Owing to more advanced equipment and algorithms, we obtained MDs containing more biological information. A comprehensive analysis and experiment proved the superiority of MDs for pathogen identification. An end-to-end network design, namely, BI-Net, based on feature fusion and 3D-CNN dramatically improved data utilization and enhanced the representative capability. By combining the advantages of the equipment and algorithms, we achieved rapid and accurate identification of infectious pathogens.

By transferring the knowledge obtained regarding common pathogens to uncommon pathogens, we accurately identified uncommon pathogens using only small sample learning, thus solving the problem that uncommon pathogens cannot be trained using data-driven methods due to small sample sizes. The recognition accuracy was 92% for both common and uncommon pathogens, with a Kappa coefficient of 0.92. 

Previous studies have mostly focused on food-borne pathogens and have involved relatively few varieties of pathogens and a small number of samples. Compared with our previous work [28,30], we have greatly expanded the category and number of pathogens in the dataset. To more closely match the complex clinical diagnostic environment, unlike previous studies that utilized bacteria from a single bodily fluid, we included up to 16 types. In addition, the number of bacterial strains was far greater than that in previous studies. Moreover, previous methods were limited in their ability to exploit information gained from MDs. By designing the end-to-end algorithm called BI-Net, the approach used in this study greatly outperformed the state-of-the-art methods. 

In addition, the spectral and morphological profiles depend on the hardware system, slide-making process, and imaging environment. These differences can cause the spectral and morphological profiles of the obtained MDs to differ, thereby reducing the accuracy of the model. To solve this problem and improve the method’s robustness, we carried out the following: 1. We calibrated the hardware system with a whiteboard as a reference to remove the hardware differences, including the focal plane. 2. We standardized the slide-making process. The thickness of the slides used in one batch was the same. Even if the brightness of the images differed due to differences in slide thickness, the image of the input model was relatively fixed since it needed to be divided by the average background to obtain the transmittance image. The background was the average spectrum of the areas of the image without bacteria. This method eliminates the differences in image brightness resulting from different slide thicknesses. 3. The imaging environment, such as light source intensity, can also affect the image brightness, but this problem was solved when we dealt with different slide thicknesses. Therefore, after our process, these differences did not affect the accuracy of the model. Furthermore, the model is highly portable. After training, it only needs to be copied to the required equipment for use without further training, significantly saving time and money.

An interesting finding was that different species of the same genus, although highly genetically similar, were less likely to be misclassified as each other. However, this result has not previously been described and might be used to identify species of the same genus that are difficult to differentiate using medical methods.

In the current study, we performed training and testing using purely cultured bacteria and obtained good results. Our method looks more complex than studies that use colonies for pathogen classification. However, our study is to lay the data and model foundation for the direct identification of pathogens that are not cultured. Next, we are working on identifying pathogens without culture based on the results of this paper.

## 5. Conclusions

This study explored species identification of individual bacteria using large-scale cell-level pathogen MDs and deep learning. We achieved rapid pathogen identification using a deep learning algorithm, BI-Net, which enhances the representation ability of features by fusing high-level semantic features and low-layer detail features. With the help of transfer learning, we also achieved high accuracy in identifying uncommon pathogens. Compared to previous studies, we further explored the data distribution patterns of infectious agents by using more advanced algorithms and more challenging data for a more comprehensive analysis, resulting in the accurate and rapid classification of individual bacteria. It was verified that using the MDs of individual bacteria after pure culture is amenable to species identification. In the follow-up study, we will try to identify single bacteria without culture based on the present data and algorithm. In fact, this is what we have done to promote the clinical application of MDs.

## Figures and Tables

**Figure 1 cells-12-00379-f001:**
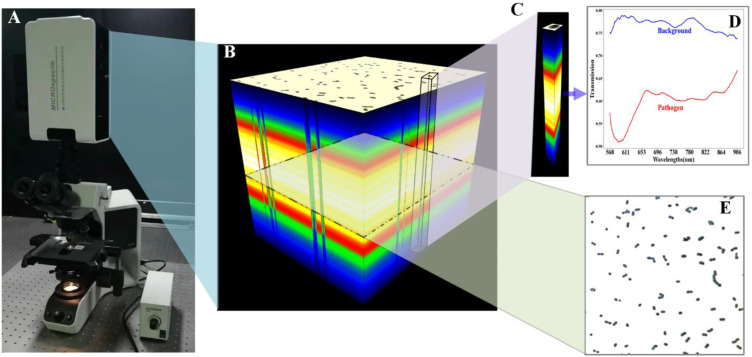
Hyperspectral microscopic imaging. (**A**) The HMI system, consisting of a microscope and a hyperspectral sensor. (**B**) An original MD, which has a dimension of 1000 × 1000 × 277. (**C**) MD of a single bacterium cropped from the original MD, which was used as an experimental sample. (**D**) Spectral profiles of bacteria and background. (**E**) An intensity image of individual bands in the original MD, reflecting the bacteria morphology.

**Figure 2 cells-12-00379-f002:**
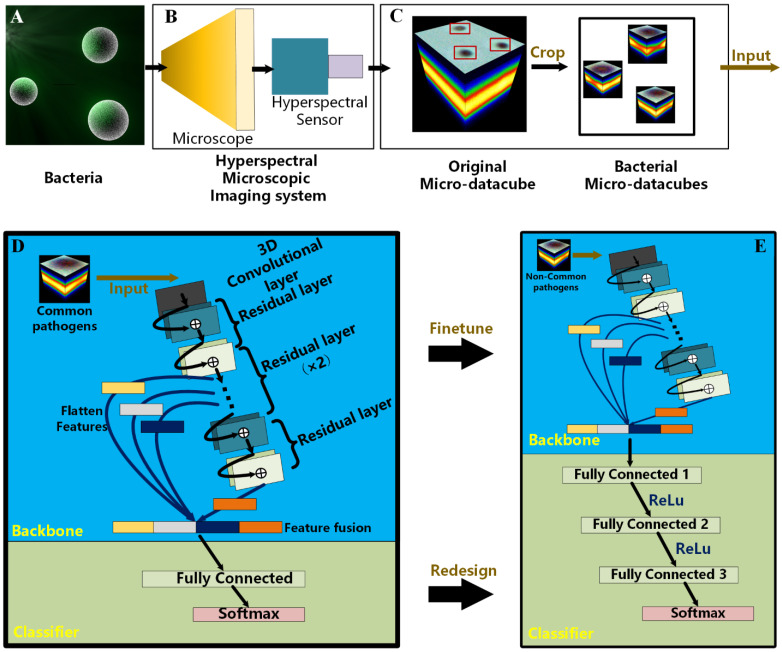
The workflow for detecting bacterial species with MDs and deep learning. (**A**) The illustration of the three-dimensional model for pathogens. (**B**) The structure of the hyperspectral microscope. (**C**) The MDs of individual bacterium fed into the BI-Net were cropped from the original MD. (**D**) The structure of the BI-Net proposed for identifying common pathogens, which can be divided into two parts: backbone and classifier. Based on the 3D-CNN, the backbone extracted the spatial–spectral features from the bacteria and fused them into a classifier. The classifier was used for bacterial identification with a fully connected layer and SoftMax. (**E**) The identification of uncommon species was based on BI-Net after transfer learning with the same structure and finetuned weights of the backbone. The classifier was redesigned and retrained.

**Figure 3 cells-12-00379-f003:**
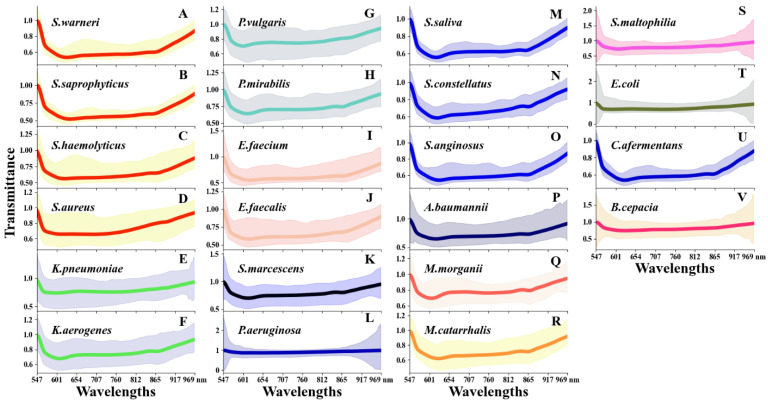
Spectral profiles of 22 species. The species in the same genus are listed in a column and visualized with the same color. The thick central curve represents the mean of the spectral profiles, and the area around the darker color represents the spectral span between the maximum and minimum values.

**Figure 4 cells-12-00379-f004:**
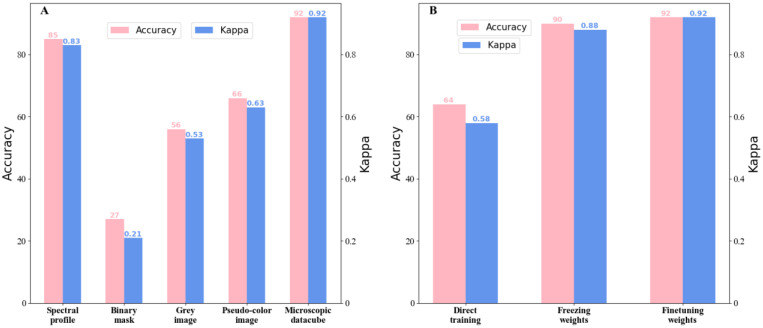
Results of comparative experiments. (**A**) Suitability analysis of MDs for identifying species. The classification results using MDs were compared with the other four data types: (1) spectral profile: the average spectral profile of an individual bacterium; (2) binary mask: containing most morphological information of the bacterium such as area, perimeter, and aspect ratio; (3) grey image: single-channel images simulating ordinary optical imaging; (4) pseudo-color image: three-channel images simulating ordinary optical imaging; and (5) microscopic datacube: containing spatial information excluding abundant spectral characteristics. (**B**) Analyzing the necessity of transfer learning. Direct training indicates that no transfer learning was used. In other words, the network was trained from scratch to identify uncommon pathogens. Freezing/finetuning weights indicate freezing/finetuning weights during transfer learning.

**Figure 5 cells-12-00379-f005:**
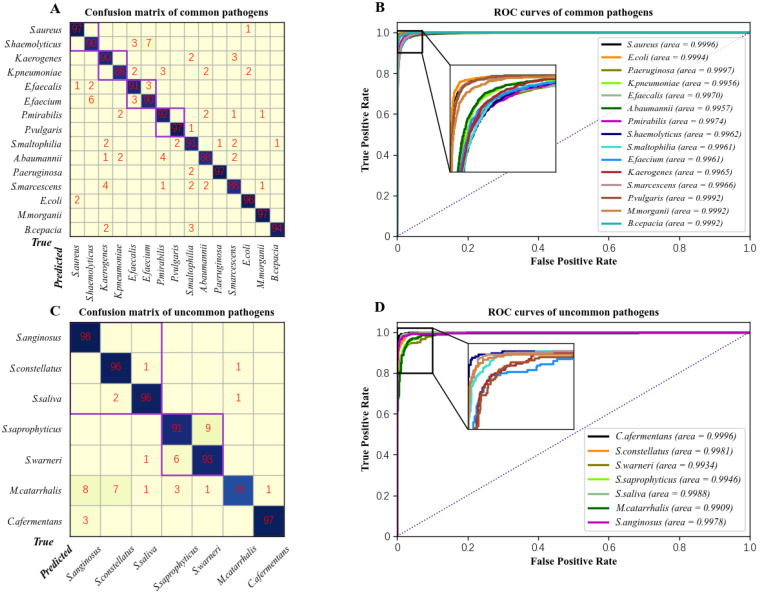
Performance of deep learning for common and uncommon pathogens. (**A**) Confusion matrix for identification of common pathogens. (**B**) ROC curve for identification of common pathogens. (**C**) Confusion matrix for identification of uncommon pathogens. (**D**) ROC curve for identification of uncommon pathogens. In the confusion matrix, species belonging to the same genus are boxed in the same rectangle.

**Figure 6 cells-12-00379-f006:**
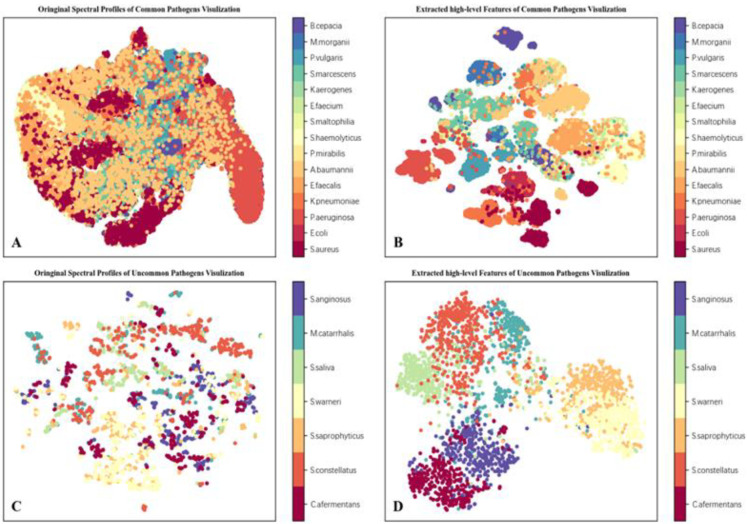
Feature visualization for pathogens. (**A**) t-SNE visualization of original spectral features for common pathogens. (**B**) t-SNE visualization of high-level features extracted by BI-Net for common pathogens. (**C**) t-SNE visualization of original spectral features for uncommon pathogens. (**D**) t-SNE visualization of high-level features extracted by BI-Net transferred learning for uncommon pathogens.

**Table 1 cells-12-00379-t001:** The super parameters.

Name	Common Pathogens	Uncommon Pathogens
Training batch size	128	16
Testing batch size	128	128
Epoch	30	40
Learning rate	10^−5^	10^−5^
Momentum	0.9	0.9

**Table 2 cells-12-00379-t002:** The identification results in the test set using different classifiers.

Classifier	Accuracy	Kappa
1D-CNN [25]	57	0.54
Fusion-Net [24]	57	0.54
ResNet [40]	70	0.65
DenseNet [43]	78	0.76
BI-Net (our)	92	0.92

## Data Availability

Data will be made available by the corresponding author on reasonable request.

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
