# Peer review of "Rapid Identification of Infectious Pathogens at the Single-Cell Level via Combining Hyperspectral Microscopic Images and Deep Learning"

_cells, 2023, doi:10.3390/cells12030379_

Round 1

Reviewer 1 Report

In this manuscript, Tao et al developed an analytical workflow for identifying the bacterial species based on their hyperspectral images. They used deep learning frameworks to leverage the spectral and morphological information from the bacteria to achieve that. The paper is clearly written, and the results are sound. I would like to ask the authors to address the following issues regarding the practical use of this method in the clinical setting. 

1. A clinical sample usually contains multiple pathogen species of bacteria that live in an environment quite different from the lab culture conditions. How transformable are the training data in real-life samples? Have the authors tried to identify bacteria from such samples?

2. The spectral and morphological profiles should depend on the imaging condition, such as the focal plane, the thickness of the coverglass, and the spectral/intensity profile of the lamp, etc. The authors should discuss how these difference may affect the accuracy of their model. Especially, to obtain high accuracy, is it necessary to train the model every time for an imaging session? If so, the cost could be high.

Minor issues:

1. Abstract: What does "tremulous" mean here?

2. The introduction should include discussion of PCR-based genomic approaches.

3. Equation 3: MDb should be SCb?

4. Figure 3: "transmissance" should be "transmittance".

5. The authors should consider making their codes open source to facilitate the application of their method.

Reviewer 2 Report

See the attached MS Word file.

Round 2

Reviewer 2 Report

The revised version of the manuscript reads much better. I must confess I am still not entirely convinced that classifying spectra of single bacteria is a smart long-term strategy for the development of diagnostic systems. However, my disagreement with the authors should not prevent them from publishing their results. It is indeed intriguing that single organisms exhibit distinguishable spectral profiles. Therefore, the authors' report may contribute to the creation of new techniques that take advantage of these spectral properties. Even if the result does not necessarily lead to improvement in diagnostics (as the authors envisioned), it may inspire other label-free biodetection work.